# Monte Carlo-Based Nanoscale Dosimetry Holds Promise for Radiopharmaceutical Therapy Involving Auger Electron Emitters

**DOI:** 10.3390/cancers16132349

**Published:** 2024-06-26

**Authors:** Ohyun Kwon, Sabrina L. V. Hoffman, Paul A. Ellison, Bryan P. Bednarz

**Affiliations:** Department of Medical Physics, University of Wisconsin School of Medicine and Public Health, Madison, WI 53705, USA; okwon25@wisc.edu (O.K.); sabrina.hoffman@wisc.edu (S.L.V.H.); paellison@wisc.edu (P.A.E.)

**Keywords:** radiopharmaceutical, Monte Carlo, Geant4-DNA, DNA damage, auger electron emitter

## Abstract

**Simple Summary:**

In contrast to external beam radiation therapy, our understanding of the dose–response relationship in radiopharmaceutical therapy (RPT) remains limited. Given that the estimation of radiation-induced deoxyribonucleic acid (DNA) damage can correlate with various biological responses, nanoscale dosimetry emerges as a critical tool in lighting this intricate relationship. In this study, we developed a novel simulation-based nanoscale dosimetry platform by identifying optimal DNA physics models and benchmarking against existing data. Leveraging this platform, we conducted a case study to estimate DNA damage, thereby enhancing our comprehension of RPT effects at nanoscales. This research serves as a foundation for future studies on how RPT affects DNA at the nano-level, potentially improving cancer treatment strategies.

**Abstract:**

Radiopharmaceutical therapy (RPT) is evolving as a promising strategy for treating cancer. As interest grows in short-range particles, like Auger electrons, understanding the dose–response relationship at the deoxyribonucleic acid (DNA) level has become essential. In this study, we used the Geant4-DNA toolkit to evaluate DNA damage caused by the Auger-electron-emitting isotope I-125. We compared the energy deposition and single strand break (SSB) yield at each base pair location in a short B-form DNA (B-DNA) geometry with existing simulation and experimental data, considering both physical direct and chemical indirect hits. Additionally, we evaluated dosimetric differences between our high-resolution B-DNA target and a previously published simple B-DNA geometry. Overall, our benchmarking results for SSB yield from I-125 decay exhibited good agreement with both simulation and experimental data. Using this simulation, we then evaluated the SSB and double strand break (DSB) yields caused by a theranostic Br-77-labeled poly ADP ribose polymerase (PARP) inhibitor radiopharmaceutical. The results indicated a predominant contribution of chemical indirect hits over physical direct hits in generating SSB and DSB. This study lays the foundation for future investigations into the nano-dosimetric properties of RPT.

## 1. Introduction

Radiopharmaceuticals are experiencing a growing presence within the field of clinical oncology, emerging as a valuable therapeutic tool for addressing an array of malignancies [1]. Radiopharmaceutical therapy (RPT) offers a distinct approach for delivering targeted radiation to widespread metastatic disease in patients by combining radioisotopes with carrier molecules that seek out tumor-specific targets. However, because the therapy is often delivered systemically, off-target exposure to normal tissues and organs will occur. The goal of any RPT treatment is to control or eliminate tumor growth while minimizing toxicity in the patient [2]. Personalized dosimetry allows for a tailored approach aimed at balancing efficacy and safety. An individual’s cancer attributes, such as tumor burden and radiation sensitivity, often assessed by the use of companion diagnostic radiopharmaceuticals, can better characterize the absorbed dose–response relationships for tumor control and adverse events [3].

Given the rising use of RPT, there is a clinical need to expand our understanding of absorbed dose–response relationships at different targets for different drugs, either delivered alone or in combination with other therapies. Currently, absorbed dose–response relationships for RPT and the radiobiological parameters that help define them are usually derived from external beam radiation therapy (EBRT) conditions. However, there are several reasons why these relationships will not properly translate to RPT. (1) First, RPT involves low dose–rate exposures that are vastly different than typical dose rates delivered in EBRT (i.e., Gy per day or week versus Gy per min or sec). As a result, cancer or other mitotically active cells undergo exposure to radiation across multiple cell cycles, with their radiation sensitivity varying throughout [4]. (2) Absorbed dose distributions in RPT are far more heterogeneous than those delivered in EBRT. Dose distributions in RPT are influenced by vasculature structure and blood perfusion due to the systemic administration of radiopharmaceuticals. It is well known that dose heterogeneity impacts response [5,6]. (3) RPT encompasses a broad spectrum of ionizing radiation that is emitted during decay with varying levels of radiation quality, which refers to the characteristics of the radiation such as particle type, energy, and linear energy transfer (LET). The impact of radiation quality at the cellular or deoxyribonucleic acid (DNA) level for a variety of these particles is poorly understood. (4) Finally, RPT introduces off-target effects, such as bystander effects and immune responses, that are different from EBRT.

Ionizing radiation primarily initiates early damage to genes or cells through the initial damage to specific segments of DNA. This type of injury arises from the intricate spatial distribution of inelastic interactions caused by individual ionizing particles within the DNA molecule or its immediate vicinity. Consequently, this damage depends on the probabilistic nature of particle interactions occurring within volumes as small as a few nanometers. Radiation damage at these scales is significantly influenced by the quality of radiation and cannot be adequately elucidated by relying solely on macroscopic measures such as absorbed dose or LET. Nanoscale dosimetry holds the potential to improve our understanding of the behavior of ionizing radiation emitted from RPT drugs at these small scales, which will ultimately improve dose–response relationships for RPT.

To better understand DNA damage caused by RPT drugs that contain short-ranged emitters, we recognize the necessity of benchmarking our work using nanoscale dosimetry. To this end, the primary focus of this paper is to compare our nanoscale dosimetry simulations with previously reported benchmarks. Specifically, we chose to compare our results against the works of Pomplun and Kandaiya et al., who explored scenarios involving a short B-form DNA (B-DNA) structure and Iodine-125 (I-125), which emits short-ranged high-LET Auger electrons [7,8]. The “B” in B-DNA stands for B-form or B-conformation of DNA. B-DNA is one of the three main and most common conformations of DNA, the others being A-DNA and Z-DNA. We also investigated the impact of using a simplified model of DNA structure in these conditions by comparing our work to the results recently reported by Thompson et al. [9]. Through these efforts, we aim to provide a robust and reliable platform for understanding the impact of RPT at the nanoscale.

## 2. Materials and Methods

### 2.1. B-DNA Description

Two geometric structures of the DNA model were benchmarked to simulate radiation-induced DNA damage (Figure 1). First, the high-resolution B-DNA structure was reconstructed using the PyMOL software (version 2.4) [10], following the model of Pomplun. This model comprises 82 nucleotides forming 41 base pairs (BP), each with a thickness of 0.34 nm and a rotation angle of 36 degrees. The generated structure closely matched the rotation angle to Charlton’s model, and the bond length differed by less than 5% compared with the values reported in the study by Chandrasekaran and Arnott [11,12]. This model is encapsulated in a virtual cylinder with a diameter of 2.4 nm and a height of 14.3 nm to account for physical indirect damage matching Pomplun’s method. The model was imported into the Geant4 framework (version 10.07), utilizing a PDB4DNA example as guidance [13,14,15,16]. Moreover, a simple B-DNA structure was generated directly in Geant4 following the approach outlined in Thompson’s study, which employed models referenced from studies by Nikjoo and Charlton [17,18]. The overall cylinder dimensions of the simple model are similar to those of the high-resolution model, with a total cylinder diameter of 2.3 nm, a nucleotide thickness of 0.34 nm, and a rotational angle of 36 degrees, resulting in an overall height of 14.3 nm. However, it features a repetitive structure of a simple sugar–phosphate backbone forming a thin U-shape.

### 2.2. Geant4-DNA Overview

#### 2.2.1. Simulation Options

Geant4-DNA was chosen for this study due to several advantageous features over other Monte Carlo simulation toolkits. One significant advantage is its ability to import various DNA structures using Protein Data Bank (PDB) files, enabling precise simulations of complex DNA geometries. Also, it offers comprehensive modeling of the physical, physicochemical, and chemical stages of radiation interactions, with extensive validation against experimental data. Geant4-DNA libraries were used to simulate ionizing radiation track structures in and around the DNA models [19,20,21,22]. Geant4-DNA offers a selection of three distinct physics constructors [23]: “G4EmDNAPhysics_option2”, “G4EmDNAPhysics_option4”, and “G4EmDNAPhysics_option6”. An overview of the electron interaction references pertinent to each of these constructors is presented in Table 1. To maintain precision, a production cut distance of 0.1 nm was implemented for secondary electrons. Moreover, after the initial physical stage lasting 10^−15^ s, we extended our study utilizing DNA chemistry options to include the pre-chemical and chemical stages, spanning up to 10^−6^ s [24,25], thereby enhancing the comprehensiveness of our approach. All simulations are performed in liquid water with a uniform density of 1 g/cm^3^.

Geant4 incorporates a radionuclide decay constructor that includes the atomic rearrangement mode (ARMflag) as delineated in the Evaluated Atomic Data Library (EADL) document [26]. Furthermore, the utilization of the deexcitation cut-off energy ignorance function facilitated the complete decay spectrum, including low energy (<1 keV) internal conversion and Auger electrons. To confirm the diverse functions and options of the framework, a validation study was executed previously. It employed the MIRDcell (version 3.12) S-value calculation program and was centered around simple cellular colony structures [27,28]. The calculations involved the determination of self- and cross-dose metrics for single-cell colonies, each characterized by a 10 μm diameter. A variety of clinical and preclinical radioisotopes were utilized in the simulations, including Yttrium-90 (Y-90), Lutetium-177 (Lu-177), Iodine-131 (I-131), and Bromine-77 (Br-77). Of particular significance, Br-77 serves as a high-LET Auger electron emitter, emitting 6–7 Auger electrons per decay [29], which imparts unique intricacies to its decay process. Overall, less than 8% and 15% differences were observed in the self-dose and cross-doses in the S-value comparison, respectively.

#### 2.2.2. Direct and Indirect Damage

The impact of ionizing radiation emitted from radioisotopes on DNA manifests through a combination of direct energy deposition and indirect chemical reactions. Central to this investigation is understanding the radiation-induced damage to the unique structural composition of the B-DNA, characterized by 41 BP locations. Each BP location consists of two DNA strands—one labeled by the radioisotope source attached at the 21 BP location and the other non-labeled. Each strand harbors a discrete base region and a sugar–phosphate region. Our model delineates these strands by the composition of atoms, each possessing a specific radius. This structural intricacy allows for individual BP locations and associated strands to be defined, making it feasible to calculate physical direct energy deposition for each BP location. Physical indirect energy deposition was also incorporated by setting a region of interest as a virtual cylinder volume, excluding the actual DNA targets, to address chemical indirect effects without relying on a chemistry option. Indirect chemical reactions were explored using the Geant4-DNA chemistry option. These reactions, arising from interactions between radiation-generated free radicals and the DNA target model, contribute significantly to the overall damage. Figure 2 illustrates the Geant4 simulation results attributed to physical direct hits (left panel) and the Brownian diffusion trajectories corresponding to chemical indirect hits (right panel).

#### 2.2.3. Single Strand Break (SSB) and Double Strand Break (DSB)

The sugar–phosphate backbone structure can be damaged, resulting in a strand break (SB) due to physical direct action or chemical indirect action. In our work, SSBs were tallied when direct energy deposition over 10 eV occurred within the corresponding atomic volume or when chemical reactions occurred within the reaction radius of the volume (Table 2) [30,31]. DSBs were categorized into correlated physical, uncorrelated physical, uncorrelated physical and chemical, and uncorrelated chemical-only DSBs based on distinct energy deposition patterns and radical interactions (Figure 3). A distance threshold for DSB, denoted as d_DSB_, of 3, 5, and 10 BP intervals was considered. Additionally, similar definitions for SSBs were applied to the semi-cylindrical Thompson geometry, incorporating physical and chemical reactions as physical reactions within each semi-cylinder at the BP location, with a threshold energy deposition value of 17.5 eV, as detailed in relevant publications [9,17,18]. Furthermore, our simulation focused solely on the interaction between radiation and matter within the DNA structure and did not account for potential biological interactions such as nucleobase damage and its role in SSB and DSB formation. This decision was made to maintain the scope of the simulation, emphasizing physical rather than biological interactions.

### 2.3. Benchmarking with Simulation Data

Our work was compared with the data presented by Pomplun [7]. This involved comparing the total energy deposition per decay within the base and sugar–phosphate regions at each BP location on the labeled strand. To determine the most appropriate G4EmDNAPhysics list, three distinct options previously described were evaluated against Pomplun’s data. Subsequently, the selected physics list was employed in simulations and compared with benchmark data. These simulations were further benchmarked against Pomplun data for both labeled and non-labeled strands. The analysis considered both direct and indirect energy deposition.

### 2.4. Benchmarking with Experimental Data

The probability of SSB induced by physical direct hits per decay was derived from the energy deposition distribution at each BP location. The results were benchmarked against experimental data as provided by Kandaiya [8], who conducted a study on SSB induced in a 41-mer oligonucleotide by physical direct hits from the decay of DNA-incorporated I-125-deoxycytidine using an experimental approach. The initial simulation solely modeled physical direct hits by eliminating chemical species to replicate the reported experimental conditions, which utilized a dimethyl sulfoxide (DMSO) environment. Note that DMSO acts as a free radical scavenger, thereby significantly reducing DNA damage attributed to indirect chemical effects. Subsequently, a second comparison was conducted against experimental data without DMSO, encompassing both physical direct and chemical indirect hits, for a comprehensive assessment of the simulation’s accuracy. Notably, only the hydroxyl free radical (˙OH) was considered for indirect events. This decision is rooted in the fact that approximately 80 percent of the cell is composed of water, rendering the reactions of ˙OH, a highly reactive transient species, particularly significant in biological damage. ˙OH stands out as one of the most potent chemical oxidizing agents and is one of the two primary reactive species resulting from water decomposition [17,32,33]. Given that ˙OH can have an impact over 2–3 times larger than other reactive species, we maintained this simplification by initially focusing solely on ˙OH to evaluate its relative impact.

### 2.5. Br-77 Example

We have previously reported on a theranostic poly-ADP ribose polymerase (PARP) inhibitor for targeted RPT of ovarian cancer in vitro and positron emission tomography (PET) imaging of healthy mice in vivo [34,35]. This RPT agent has also been investigated for radiotherapeutic efficacy in preclinical murine models of prostate cancer [36]. Using our simulation-based nano-dosimetric platform, we modeled a scenario involving DNA damage resulting from the decay of a Br-77-labeled PARP inhibitor bound to the active site of a PARP1 protein that is, in turn, bound to DNA. To generate the scenario, we incorporated three models obtained from the PDB. The models utilized were based on a portion of PARP1 bound to SSB DNA (PDB ID: 2n8a) [37], a distinct portion of PARP1 bound to the PARP inhibitor rucaparib (PDB ID: 6vkk) [38], and the entire PARP1 protein bound to DSB DNA (PDB ID: 4dqy) [39]. These models were used to determine the distance between the PARP inhibitor and the DNA and to locate the Br-77 source within the protein geometry (Figure 4a). The results indicated that the Br-77 radionuclide payload ends up 5 nm away from the DNA when incorporated into a rucaparib analogue bound to the active site of a DNA-bound PARP1 protein. Subsequently, these models were utilized to estimate the occurrence of SSB and DSB, resulting in the 2n8a model from the Br-77 radionuclide attachment (Figure 4b–d). Our study was performed invoking different d_DSB_, spanning distances of 3, 5, and 10 BP intervals, and accounting for multiple chemical species, including ˙OH, hydrogen radical (H˙), and hydrogen peroxide (H_2_O_2_). We also considered different source distances at 1 nm (proximate to DNA) and 5 nm (PARP inhibitor binding site distance to DNA), and the contributions to SSB and DSB from both physical direct and chemical indirect hits were evaluated within this context.

## 3. Results

### 3.1. Benchmarking Simulation Data Results

Figure 5 presents a comparison of the energy deposited in different base and sugar–phosphate regions for each of the three DNA physics lists in relation to MOCA8 simulation data reported by Pomplun [7]. In this simulation, option 4 was identified as having the best agreement between the DNA physics setting and Pomplun’s data. We then utilized this option to assess mean energy deposition in physical direct and indirect hits at base and sugar–phosphate positions on both source-labeled and non-labeled strands (Figure 6). The results showed a minimum percent difference ranging from 1.4% to 6.5% at peak (near-source) positions, while notably larger differences, approximately 70%, were observed at the edges. While the overall trends in the graphs exhibit similarities, the observed significant deviation in results is likely due to variations in simulation conditions, most notably differences in physical interaction cross-sections. Additionally, other contributing factors, including distinctions in geometry, emitted particles from decay, and the electron transport physics within the simulation, play a role in these observed deviations.

### 3.2. Benchmarking Experimental Data Results

The probability of SSB per decay at various BP locations was calculated using a computationally benchmarked simulation against Pomplun. Subsequently, it was compared with experimental data provided by Kandaiya, alongside other references employing the calculation, experiment, and simulation for similar approaches involving a B-DNA structure and the I-125 radionuclide [40,41] (Figure 7). Consistent trends between simulations and experimental data were observed in three evaluations. The first comparison involved physical direct hits simulated in Geant4-DNA and experimental data with DMSO reported by Kandaiya, showing an average difference of 28.2 ± 18.3% in Figure 7 (top). The second comparison evaluated physical direct energy depositions and chemical indirect reactions, focusing exclusively on ˙OH free radicals. Geant4-DNA chemistry option 3, using the independent reaction times (IRT) technique, was compared with experimental data without DMSO from Kandaiya, resulting in an average difference of 28.9 ± 18.5% in Figure 7 (middle) [25].

Finally, a simple B-DNA structure was utilized, applying physical direct hits with a threshold energy of 17.5 eV. This approach assumes the inclusion of both physical and chemical interactions within a large semi-cylindrical volume using only physical hits with a relatively high threshold energy. The outcomes were then compared with high-resolution results and corresponding experimental data in Figure 7 (bottom). As a result, the high-resolution B-DNA structure demonstrated superior agreement with the experimental dataset.

### 3.3. Br-77 Results

The total energy deposited by emitted radiation into the DNA model bounding box in the Br-77 source decay scenario at distances of 1 nm and 5 nm from the DNA was 0.182 and 0.013 keV/decay, respectively. Contributions to SSB from different chemical species were as follows: 70.2% from all species, with approximately 70% from ˙OH, 30% from H˙, and 0% from H_2_O_2_. Moreover, contributions to DSB from chemical-induced reactions (SSB_p_ + SSB_c_ and SSB_c_ + SSB_c_) were as follows: 91.3% from all species, with ˙OH having the most significant impact, followed by H˙. H_2_O_2_ did not contribute.

Considering different DSB threshold distances, when the BP interval was decreased from 10 to 5 and 3, the SSB per decay value increased by 35.2% and 48.4%, respectively. Conversely, the DSB per decay value decreased by 60.9% and 85.0%, respectively. Table 3 summarizes results pertaining to the 10 BP DSB threshold interval with all three chemical species at 1 nm and 5 nm radionuclide-DNA distances. In general, the distribution of contributions for SSB and DSB was similar at both distances of 1 nm and 5 nm, with 70% and 90% from chemical-induced damage for SSB and DSB, respectively.

## 4. Discussion

This paper presents a comparison of Geant4-DNA simulation data to simulation and experimental benchmarks of radioactive decay on or near DNA. Data acquired from Pomplun and Kandaiya et al. were used for benchmarking [7,8]. In summary, the percent difference of energy deposition per decay against Pomplun simulation data ranges from 1.4% to 6.5% near the source but increases to 70% at the edges. For Kandaiya’s experimental data, the average percent difference of SSB per decay with and without DMSO is 28.2 ± 18.3% and 28.9 ± 18.5%, respectively. In both cases, the impact of I-125, a source of short-ranged Auger electrons, was evaluated when labeled with B-DNA. Additionally, we sought to assess the implications of employing a simplified model for DNA structure within these conditions by drawing comparisons with the recent findings reported by Thompson et al. Our objective for this benchmarking work is to establish an accurate and robust computational framework for helping to elucidate the ramifications of RPT at the nanoscale. Finally, we presented a case study where we utilized this framework to evaluate the impact of a Br-77-labeled PARP inhibitor attached to a computational model of DNA.

The Geant4-DNA calculations yielded a mean energy deposition profile that demonstrated reasonable agreement with the data presented by Pomplun. Notably, this agreement was particularly pronounced in regions closer to the site of decay. However, disparities emerged at locations distant from the decay site (Figure 5 and Figure 6), primarily attributable to variations in the cross-section environment—a factor previously acknowledged by Pomplun [7]. Specifically, Pomplun observed a tendency for energy deposition sizes to shift towards larger values along the particle tracks within the medium when considering water vapor rather than liquid water. Furthermore, differences in the software employed for B-DNA modeling contributed to the observed discrepancies. While we utilized PyMOL to generate a high-resolution B-DNA structure, Pomplun employed DNAMOD. A noteworthy distinction lies in the construction of the thymine methyl group, where our study individually modeled carbon and hydrogen atoms, in contrast with Pomplun’s approach of modeling them as a single entity. Moreover, variations in electron emission per decay further influenced the comparison. Our study identified 25.8 electrons emitted per I-125 decay in Geant4-DNA, in contrast to 21.2 electrons reported by Pomplun and 23 electrons reported by the IAEA [42]. Similarly, for Br-77 decay, Geant4-DNA yielded 7.7 electrons emitted per decay, showing a different Auger electron emission spectrum compared with MIRDcell (6.6 per decay) and Kassis (6.7 per decay) data [27,43]. In detail, the comparison revealed approximately a 20% deviation in electron emission below the 1 keV energy range and less than a 5% deviation above 1 keV. Lastly, differences in electron transport track structure codes between Geant4-DNA and MOCA8 introduced another layer of variation. Even within the same code environment, diverse transport models yielded disparate results, as evidenced in Table 1 and Figure 5. In particular, there exists a 30-year gap in the development timeline between MOCA8 and Geant4-DNA, potentially contributing to the observed discrepancies.

Much better agreement was achieved when comparing the probability of SSB induced by physical direct hits alone and SSB induced by physical direct and chemical indirect hits to the experimental results of Kandaiya in Figure 7 (top and middle, respectively). This comparison allowed us to evaluate the accuracy of our combined physical and chemical models. Similar to the comparison with the Pomplun results, the agreement between Geant-DNA and experiments lessened at greater distances from the source. There could be several reasons for this disagreement. First, the selection of chemical species inducing DNA damage in the simulation did not precisely align with the experimental conditions. In our study, we employed only ˙OH free radicals, recognized as the most influential chemical species for radiation-induced indirect DNA damage, for initial benchmarking to maintain simplicity. In contrast, Kandaiya’s experimental work likely reflected various radiation-induced free radicals, including ˙OH, H˙, and H_2_O_2_, with corresponding proportions in the solution. Secondly, differences in the diffusion and reaction times of the generated chemical species likely existed between our study and Kandaiya’s work. We employed the chem6-based DNA chemistry option 3, validated with a specific experimental setting spanning 1 ps to 1 μs. However, the authors of the chem6 example acknowledged that adapting these factors may be necessary for varying experimental setups [25]. Kandaiya’s experiments reflected actual radical diffusion and reaction times. Additionally, there were variations in reaction cross-sections. Our study exclusively employed liquid water for all diffusion and reactions, potentially inducing differences when compared with experiments that will account for the actual reaction cross-sections, including DNA molecules.

We want to draw particular attention to Figure 7 (bottom), which we believe is an important finding presented in this work. In addition to our high-resolution B-DNA model, we also modeled B-DNA using a simplified model that was presented recently by Thompson et al. When comparing the results to experimental data, the high-resolution model exhibits markedly better agreement with experimental data at BP locations ranging from 0 to 9 when compared with the simple model. It is important to note that the simple model did not incorporate the chemical stage due to its relatively simplified and bulky target volume. Instead, it only utilized a threshold energy (set at 17.5 eV), either statically or dynamically, to account for SSBs resulting exclusively from physical hits. These limitations should be considered when using the simple B-DNA structure for nanodosimetry calculations.

Lastly, the findings from the Br-77 example present detailed observations of SSB and DSB (see Table 3). The chemical-induced damage accounted for approximately 70% of SSBs and about 90% of DSBs for both 1 nm and 5 nm radionuclide-to-DNA distances, similar to the cases described in previous experimental and simulation studies in which cells were irradiated with different LETs [44,45,46]. Auger electrons, characterized as high-LET particles, were expected to have a lower percentage of chemical-induced damage contributions in the 1 nm distance scenario compared with the 5 nm distance case. However, it is noteworthy that the percentages of chemical-induced damage contributions were nearly identical in both scenarios, a trend observed up to the 0.1 μm distance. While this outcome may be influenced by the small DNA volume, limiting opportunities for sequential damage from densely ionizing particle tracks, it also underscores the fact that this study solely accounts for very early DNA damage and does not encompass DNA repair stages, which could accumulate easily reparable damage during the simulation [47]. These considerations should be recognized as potential limitations of this work.

Despite the comprehensive analysis, several limitations that we found must be acknowledged. First, our study did not account for the biological repair stage, which could impact the overall DNA damage assessment. Second, we only used selected major chemical species, such as ˙OH free radicals, which differ from the more complex mix present in reality. Third, the diffusion time of these chemical species was limited to within 1 µs, potentially overlooking longer-term interactions. Lastly, all interaction cross-sections were assumed to be within homogeneous liquid water, which may not fully represent the actual biological environment where DNA is found.

Nonetheless, our study represents a computational framework that aims to advance the exploration of dose–response relationships through Monte Carlo simulations at the nanoscale. In the context of DNA damage induced by Auger electron emitters, the results generated through comparisons with previous data involving both physical and chemical processes help validate our methods. In future work, we plan on investigating the nanodosimetric impact of other high-LET and low-LET particles, including α- and β-emitters. Therefore, we anticipate that this framework holds promise to have an impact on several research endeavors involving RPT.

## 5. Conclusions

This study successfully benchmarked the DNA damage model employing the Monte Carlo code Geant4-DNA, demonstrating reasonable agreement with simulation and experimental data. Leveraging this benchmarking, we applied the PARP inhibitor example to estimate the intricate DNA damage induced by the Br-77 radionuclide. Our work lays the foundation for forthcoming nanoscale DNA damage modeling, facilitating comprehensive dose–response investigations in this domain.

## Figures and Tables

**Figure 1 cancers-16-02349-f001:**
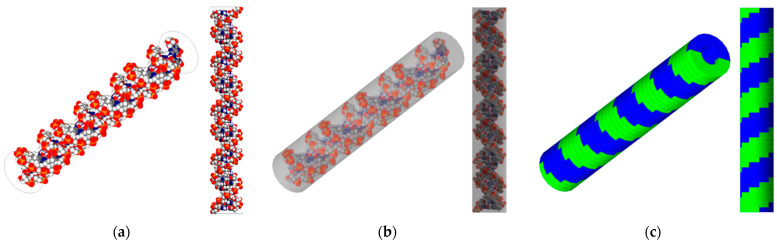
DNA target models used in this work. The left panel (**a**) depicts a high-resolution representation of an 82-nucleotide segment of the B-DNA double helix (white: hydrogen, red: oxygen, blue: nitrogen), and (**b**) shows the B-DNA with a bounding box, benchmarking Pomplun et al. [7]. The right panel (**c**) features simplified semi-cylindrical geometries representing a double-strand (one green, one blue) B-DNA backbone structure, benchmarking Thompson et al. [9].

**Figure 2 cancers-16-02349-f002:**
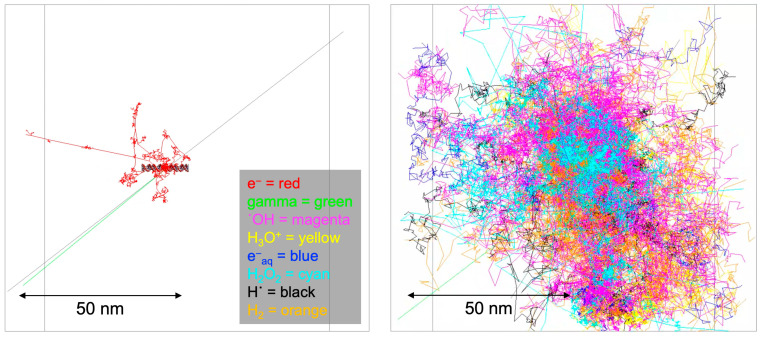
Geant4-DNA physics and chemistry stage visualization. The physics stage (**left** panel) has shown emitted particle tracks from the I-125 radioisotope for 1 decay, and the chemistry stage (**right** panel) has shown generated radical diffusion tracks by Brownian motion.

**Figure 3 cancers-16-02349-f003:**
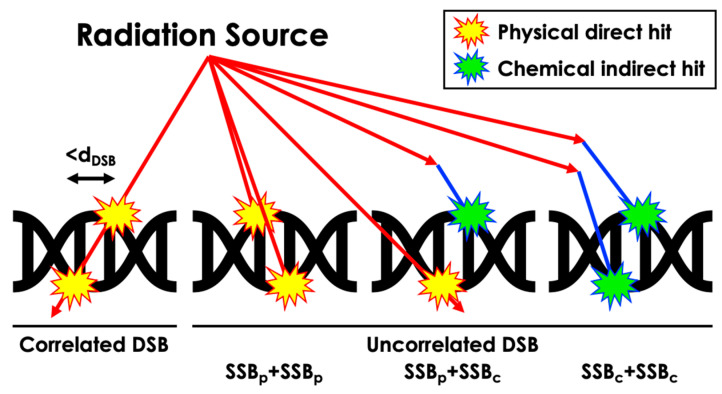
Schematic representation of the computation process for different DSB scenarios resulting from diverse radiation-induced damage types.

**Figure 4 cancers-16-02349-f004:**
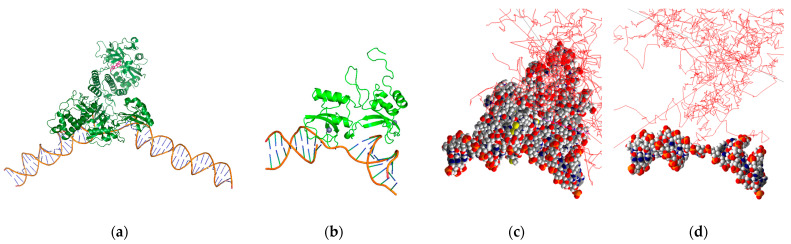
PARP1-bound SSB DNA with Br-77-labeled PARP inhibitor rucaparib (pink molecule) (**a**), PARP1 F1F2 domains in complex with SSB DNA geometries in PyMOL (**b**) and Geant4-DNA (**c**), and the DNA portion alone in Geant4-DNA (red line: electron tracks) (**d**).

**Figure 5 cancers-16-02349-f005:**
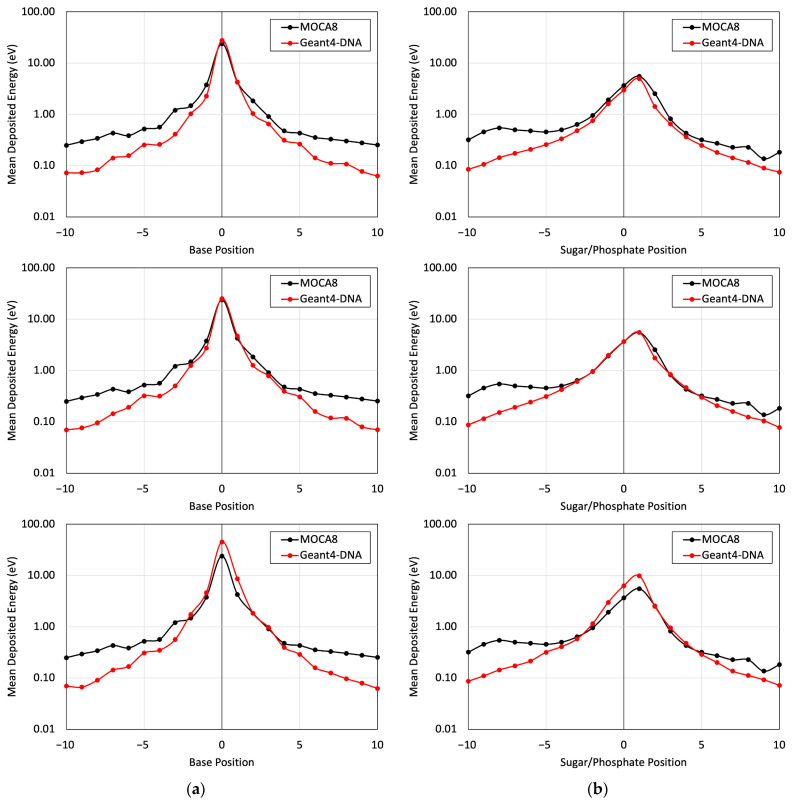
The mean energy deposition per decay from the I-125 radionuclide at the 0 position was evaluated for base positions (**a**) and sugar–phosphate positions (**b**) on the labeled strand. This assessment considered physical direct hits and utilized Geant4-DNA Physics options 2, 4, and 6 on (top), (middle), and (bottom), respectively.

**Figure 6 cancers-16-02349-f006:**
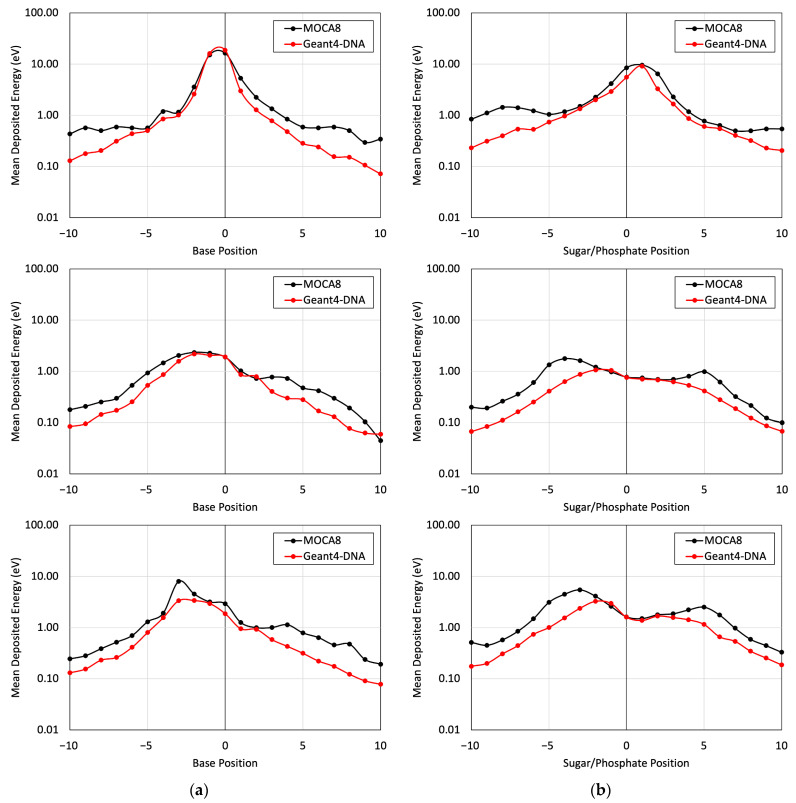
The mean energy deposition per decay from the I-125 radionuclide was evaluated for base positions (**a**) and sugar–phosphate positions (**b**). This assessment utilized Geant4-DNA Physics option 4 alone, considering (top) physical indirect hits on the labeled strand, (middle) physical direct hits on the non-labeled strand, and (bottom) physical indirect hits on the non-labeled strand.

**Figure 7 cancers-16-02349-f007:**
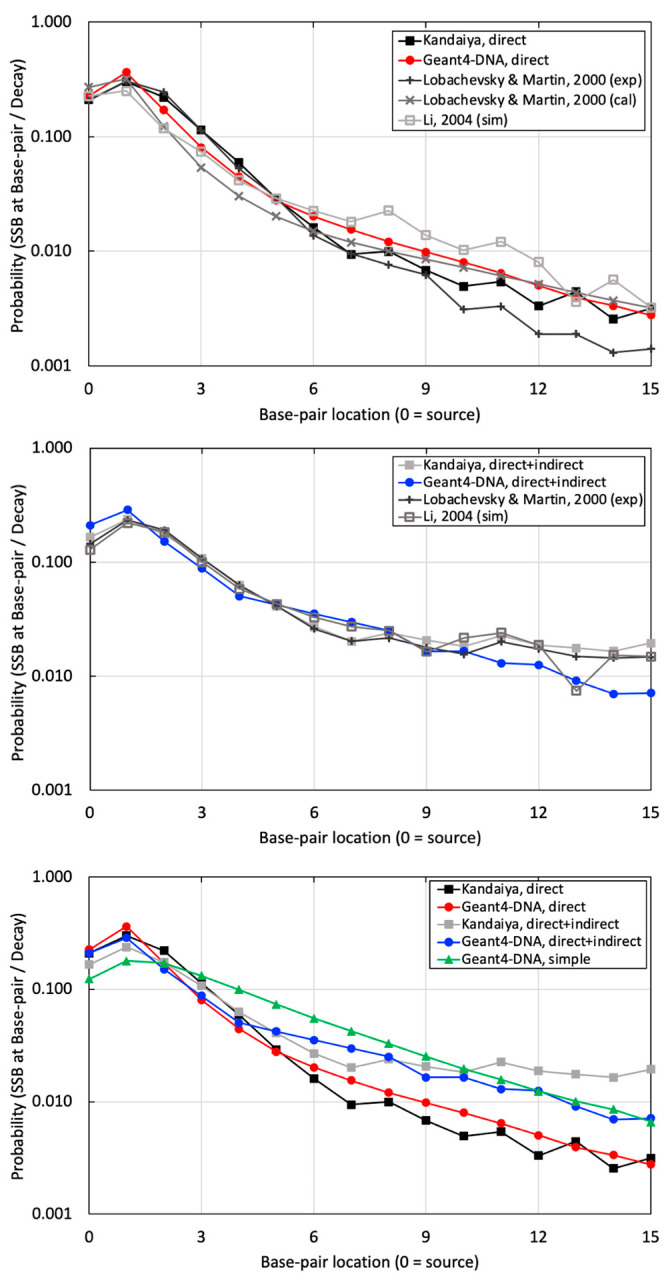
The probabilities of SSB occurring per decay at sugar–phosphate regions on the radionuclide source labeled strand, with (top) illustrating the results of SSB induced by physical direct energy deposition from our study with other references, (middle) depicting the results of SSB induced by a combination of physical direct energy deposition and chemical indirect reactions from our study with other references, and (bottom) demonstrating a comparison between the high-resolution and simplified B-DNA structure from our study against the corresponding benchmarking experimental data by Kandaiya et al. [8].

**Table 1 cancers-16-02349-t001:** Summary of the recommended Geant4-DNA physics lists detailing the applicable electron energy range and the cross-section models for each interaction [22].

	G4EmDNAPhysics		
	Option 2 *	Option 4	Option 6
Elastic scattering	Partial-wave analysis (7.4 eV–1 MeV)	Uehara screened Rutherford formula (9 eV–10 keV)	Independent Atom Method (11 eV–256 keV)
Ionization	Emfietzoglou model with Born corrections (11 eV–1 MeV)	Kyriakou-Emfietzoglou model (10 eV–10 keV)	Bethe model (11 eV–256 keV)
Excitation	Emfietzoglou model with Born corrections (9 eV–1 MeV)	Kyriakou-Emfietzoglou model (8 eV–10 keV)	Dingfelder model (11 eV–256 keV)
Cut-off energy	7.4 eV	10 eV	11 eV

* Sanche data (2 eV–100 eV) and Melton data (4 eV–13 eV) were applied for vibration excitation and molecular attachment, respectively.

**Table 2 cancers-16-02349-t002:** Radius of DNA atoms and hydroxyl free radical reaction to DNA components [30,31].

Atom	Atomic Radius (nm)	Reaction	Reaction Radius (nm)
Hydrogen	0.120	˙OH + Adenine	0.372
Carbon	0.170	˙OH + Cytosine	0.366
Nitrogen	0.150	˙OH + Guanine	0.553
Oxygen	0.140	˙OH + Thymine	0.384
Phosphorus	0.190	˙OH + Sugar	0.108

**Table 3 cancers-16-02349-t003:** Analysis of PARP inhibitor geometry with a Br-77 source at different distances.

	1 nm	5 nm
BP interval	10	
Chemical species	˙OH + H˙ + H_2_O_2_	
Energy deposition in the bounding box	0.182 keV/decay	0.013 keV/decay
SSB_total_/decay	0.728	0.049
SSB_p_/decay	0.217 (30%) *	0.014 (29%)
SSB_c_/decay	0.511 (70%)	0.035 (71%)
DSB_total_/decay	0.207	0.003
DSB_corr_/decay	0.002 (1%)	0.00004 (2%)
DSB_uncorr,(p+p)_/decay	0.016 (8%)	0.0002 (6%)
DSB_uncorr,(p+c)_/decay	0.089 (43%)	0.0012 (45%)
DSB_uncorr,(c+c)_/decay	0.100 (48%)	0.0013 (47%)

* % difference per corresponding total value. p: physical hits, c: chemical hits, corr: correlated hits, uncorr: uncorrelated hits.

## Data Availability

Data are contained within the article.

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
