# Peer review of "Monte Carlo-Based Nanoscale Dosimetry Holds Promise for Radiopharmaceutical Therapy Involving Auger Electron Emitters"

_cancers, 2024, doi:10.3390/cancers16132349_

Round 1

Reviewer 1 Report

Comments and Suggestions for Authors

Manuscript ID: Cancers-3060553

Title: Monte Carlo-based Nanoscale Dosimetry Holds Promise for Radiopharmaceutical Therapy Involving Auger Electron Emitters

After consideration of the manuscript, I have some comments as follow.

1-      The abstract is too long and requires a shortened.

2-      The aim of the study in the abstract and introduction are different (In the introduction section focused on the nanoscale, but in the abstract focused on both micro- and nano-scales).

3-      What is the full term for B-DNA, please write all abbreviations when they are used for the first time in the article.

4-      In the Materials and Methods section:

-          In Table 1, provide references and sources for obtaining the information.

-          In Figure 1, a copyright agreement for the mentioned reference is needed.

-          The resolutions of Figures 5, 6, and 7 need improvement.

-          The laboratory name and location of the work are not clear.

-          Please identify the reason for using Geant4 instead of other Mone Carlo simulation methods.

-          The authors emahasis for using high LET radiation, but they have not identified which type of radiations they used.

5-      In the discussion section, a comparison between the results with the other related literature is needed. Also, in the last paragraph provide some limitations of the study.

 I recommended that the paper need to a minor revision.

Reviewer 2 Report

Comments and Suggestions for Authors

This paper titled "Monte Carlo-based Nanoscale Dosimetry Holds Promise for Radiopharmaceutical Therapy Involving Auger Electron Emitters," written by Kwon et al., is well-constructed and discusses a critical question in the field. 

As mentioned by the authors, RPT has become a promising strategy for cancer therapy these years as more and more targets with potential binders are available.  The assessment of microdosimetry and how to interpret the RT-induced DNA damage is important for predicting therapeutic outcomes.  Overall, the paper is well-written and provides many valuable insights.

Comments and questions:

1. The resolution of the insert in Figure 2 can be further improved to help understanding. 

2. It may be a naive question. Why did you use a 42 bp B-form DNA in your study? Why not other numbers? 

3.  The study simulates the situation using Auger emitter I-125 and b+ emitter Br-77 as the radiation sources. Also, the authors considered the formation of free radicals, which are more important in causing DNA damage when low LET radiation is used for treatment. However, could you also comment on beta- and alpha-emitters? Since the free radicals play a less important role in causing DNA damage, especially when alpha emitters are used in RPT. 

4. In Result 3.1, the authors mentioned, "In this simulation, option 4 was identified as the best agreement DNA physics setting with Pomplun’s data." However, option 2 also showed pretty high agreement with previous data. Or were the authors more focused on the -5 to +5 position?

5. As we know, the nearby SSB can be viewed as DSB; how do you define DSB in your study? Only resulted from a single hit?   
